# Similar effects of fatigue induced by a repetitive pointing task on local and remote light touch and pain perception in men and women

Jason Bouffard[1,2¤a¤b]*, Zachary Weber[1,2], Lyndsey Pearsall[1,2], Kim Emery[1,2], Julie N. Côté[1,2]

1 Department of Kinesiology and Physical Education, McGill University, Montreal, Quebec, Canada,
2 Occupational Biomechanics and Ergonomics Laboratory, Michael Feil and Ted Oberfeld/CRIR Research Centre, Jewish Rehabilitation Hospital, Laval, Quebec, Canada

¤a Current address: Centre for Interdisciplinary Research in Rehabilitation and Social Integration (CIRRIS), Quebec City, Quebec, Canada
¤b Current address: Department of Kinesiology, Université Laval, Quebec City, Quebec, Canada
* Jason.bouffard@kin.ulaval.ca

**Data Availability Statement:** All relevant data are within the manuscript and its Supporting Information files.

## Abstract

### Background

Women involved in repetitive, fatiguing, jobs develop more neck and/or shoulder musculo-skeletal disorders than men. Sex differences in the pain response to exercise could contribute to the higher prevalence of neck/shoulder musculoskeletal disorders in women. The **objective** of this study was to assess sex differences in pain sensitivity following a fatiguing upper limb task. Relationships between measures of fatigue and of the sensitivity to nociceptive and to non-nociceptive stimulations were also explored.

### Methods

Thirty healthy adults (15 women) performed a fatiguing repetitive pointing task with their dominant arm. Upper limb electromyography was recorded from the dominant upper trapezius, anterior deltoid and bicep brachii and from the contralateral tibialis anterior. Before and immediately after the repetitive pointing task, pressure pain and light touch sensitivity thresholds were measured over the same muscles.

### Results

Electromyographic signs of fatigue were observed only in the anterior deltoid and biceps brachii muscles. Pressure pain thresholds over both muscles increased slightly (effect size $\leq$ 0.34), but no changes occurred over the upper trapezius and the tibialis anterior. Light touch thresholds increased moderately to importantly after the repetitive pointing task over all four muscles (effect sizes = 0.58 to 0.87). No sex differences were observed in any sensory variable. Moreover, no or weak correlations (r = -0.27 to 0.39) were observed between electromyographical signs of fatigue, light touch threshold and pressure pain threshold variables.

**Funding:** The project is funded by a Natural Sciences and Engineering Research Council grant (JNC, #RGPIN-2015-0511, https://www.nserc-crsng.gc.ca/index_eng.asp). JB was funded by the Canadian Institutes for Health Research (#MFE-146666). The funders had no role in study design, data collection and analysis, decision to publish, or preparation of the manuscript.

**Competing interests:** The authors have declared that no competing interests exist.

## Conclusions

We observed sex-independent effects of a repetitive upper limb task on the sensitivity to painful and to nonpainful stimuli. Moreover, the hypoalgesia induced by the repetitive pointing task was weak and localized, and did not directly correlate with the induced muscle fatigue. Results suggest that fatigue-related changes in the sensitivity to noxious and innocuous stimuli could not explain women's greater prevalence of neck/shoulder musculoskeletal disorders.

## Introduction

Neck and/or shoulder musculoskeletal disorders (nsMSD) affect 40% of the workforce and can lead to long-term disability and significant societal costs [1, 2]. Jobs requiring repetitive and fatiguing upper limb movements are linked to nsMSD [3]. Thus, some authors have hypothesized that muscle fatigue may be central to the nsMSD physiopathology [4–6].

However, the link between fatigue, pain and nsMSD is unclear. While many studies have shown increased spontaneous pain during fatiguing exercises (reviewed in [7]), evidence of exercise-induced decreases in pain, or hypoalgesia (EIH), is also frequent (reviewed in [8]). Exercise induced hypoalgesia has been observed following aerobic, dynamic and isometric exercises both in the muscle groups exercised and in more distant sites [8]. Muscle fatigue may play a crucial role in EIH, as low to moderate intensity contractions maintained until exhaustion appear to be more effective at inhibiting pain than stronger, shorter duration, contractions [8, 9]. However, few studies assessed concurrently EIH and objective signs of fatigue to assess their interrelationship. Electromyographic (EMG) signs of fatigue, such as a decreased median frequency and an increased amplitude [10], are particularly appealing as they could allow assessment of the relationships between sensory and motor changes during a fatiguing task on a muscle-by-muscle basis. For instance, Tse et al. (2016) showed that EMG signs of muscle fatigue were present in only 4 (i.e. anterior deltoid, posterior deltoid, latissimus dorsi and serratus anterior) out of 14 shoulder muscles assessed during an arm elevation task [11], which illustrates the spatial sensitivity of the method.

Although the mechanisms of EIH remain unclear, both opioid and non-opioid pain inhibitory systems appear to be involved. Indeed, naloxone (an opioid receptor antagonist) decreases only partly and inconsistently hypoalgesia (reviewed in [12]). Other biological (e.g. serotonergic, immune and autonomic nervous system) and psychosocial factors can also interact to modulate the effects of exercises on pain sensitivity, either positively or negatively (e.g. balance between proinflammatory and anti-inflammatory interleukins) [13, 14]. Exercise-induced sensory inhibition is, however, not specific to painful stimuli. Indeed, Han et al. (2015) showed that cutaneous acuity to non-painful stimuli is decreased following sustained isometric biceps or quadriceps contractions [15]. However, no studies concurrently assessed sensitivity changes to noxious and innocuous stimuli. It is still not clear if noxious and innocuous sensory changes following motor activities are related, and therefore share some physiological mechanisms.

Individual factors, such as the sex of an individual, can also influence the sensory response to motor activities. As previously stated, the prevalence of nsMSD among individuals performing work-related upper limb repetitive activities is greater for women than for men [16]. However, the sex-specific biological factors underlying these epidemiological differences, which may or may not involve sex differences in fatigue and pain mechanisms, are still unclear [5]. Surprisingly, women have been shown to display greater EIH following fatiguing single joint

tasks than men in some studies [17, 18]. Yet, sex-independent pain inhibition [19, 20], and even greater hypoalgesia in men than in women were also observed [21]. Han et al. (2015) also found some sex differences in the effects of exercise on innocuous sensory sensitivity, as cutaneous acuity decreased more with fatigue induced by isometric contractions in women than in men [15]. Regarding motor adaptations to fatigue, previous studies have shown sex differences in fatigability and in associated performance characteristics [22–24]. Collectively, these studies highlight the complex sensorimotor interactions that are present during fatiguing motor tasks in men and women. Nevertheless, there is still a lack of studies evaluating simultaneously motor signs of fatigue and the impacts of repetitive exercises on noxious and innocuous sensory systems to characterize the sex-specific interrelationship between these factors.

Our **main objective** was to assess sex differences in light touch (LTT) and in pressure pain thresholds (PPT), and EMG before and after a fatiguing multi-joint repetitive pointing task (RPT). Furthermore, to discriminate between local and remote impacts of the RPT on the sensory and motor systems, measures were collected over upper (local) and lower (remote) limb muscles. We expected an increase in sensory thresholds, in both local and remote sites, following the RPT. Regarding EMG signs of fatigue, changes were only expected in upper limb muscles. Although these changes were expected in both sexes, we hypothesized that they would be larger in women than in men. Finally, we expected that participants showing greater EMG-based muscle fatigue would experience greater EIH **(Objective 2)** and that there would be correlations between changes in PPT and changes in LTT **(Objective 3)**.

## Materials and methods

### Participants

Fifteen men (age = 26.7 ± 6.7 years; height = 176.4 ± 7.1 cm; weight = 73.3 ± 10.8 kg; body mass index = 24.6 ± 3.9 kg/m$^2$) and 15 women (mean age: 26.3 ± 7.3 years; mean height = 168.6 ± 6.8 cm; weight = 67.3 ± 9.0 kg; body mass index = 21.8 ± 2.1 kg/m$^2$) participated in this study. Men and women were significantly different for height (p < 0.001) and body mass index (p = 0.041), but not for age (p = 0.857) or weight (p = 0.152). All participants were recruited by the research team from the institutional social network and were free of neurological and musculoskeletal injuries, pain, cardiovascular diagnoses and any other general health concern, as assessed using the Physical Activity Readiness Questionnaire (Par-Q) [25]. All participants were right-handed even though this was not an inclusion criterion. Women were not screened for contraceptive medication and were not tested at a certain point during their menstrual cycle given the inconclusive evidence regarding the effects of such factors on fatigability and EIH [26, 27]. Ethical approval was received from the Research Ethics Board of the Centre for Interdisciplinary Research in Rehabilitation (CRIR) of Greater Montreal. All participants gave their written consent before their participation to the study, which was conducted in conformity with the Declaration of Helsinski.

### Experimental protocol

Participants completed a RPT using their dominant arm until they reached a perceived fatigued state of 8/10 on the Borg CR-10 scale [24]. The protocol was similar to that used in previous studies [25]. The RPT has been extensively studied for its effects on the motor system and is known to induce EMG and kinematic changes as well as a decrease in maximal force capacity [22, 24, 28, 29], which are all indicators of fatigue. Before and after the RPT, participants underwent PPT and LTT measurements, whereas EMG data was collected during the last 30 seconds of each minute during the RPT. The study design, with testing of sensory thresholds before and after a fatiguing motor task, is consistent with that used in previous

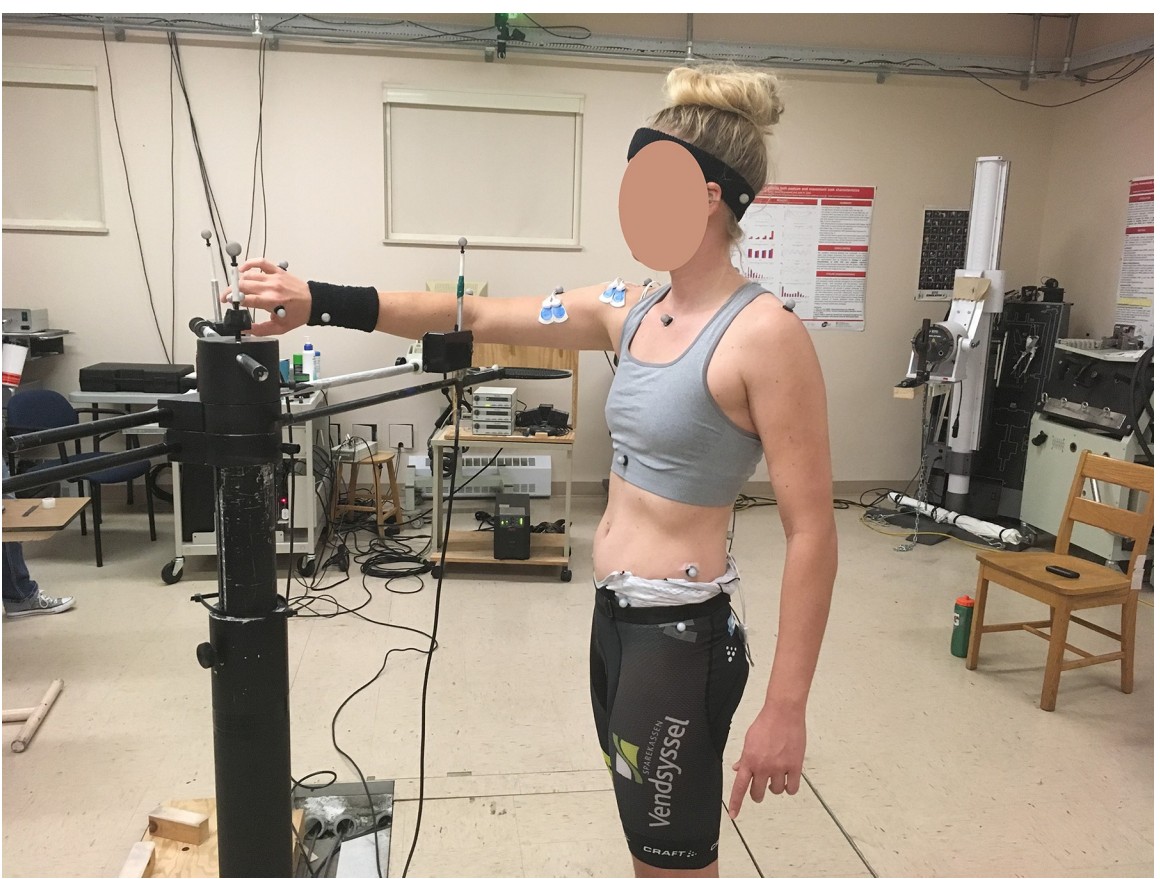

**Fig 1. Experimental set-up for the repetitive pointing task.** Note that in the present study, kinematics and pectoralis major EMG data were not collected.

studies evaluating sex differences in EIH (e.g. [19, 21]). All measurements were taken on the same four muscles. Three of them were local to the site of fatigue: the moving arm's upper trapezius (UT), anterior deltoid (AD) and bicep brachii (BB). The fourth tested muscle was remote: the contralateral tibialis anterior (TA) [30]. All measurements were carried out in a quiet room and administered by the same examiner.

During the RPT, participants stood on a area marked on the floor with tape, which was adjusted to each participant's comfortable standing position (Fig 1). Their non-dominant arm hung relaxed, close to their body. The RPT was conducted using two touch-sensitive targets (length 6 cm, radius 0.5 cm, Quantum Research Group Ltd) positioned in front of the participant's body midline at their respective shoulder height. The proximal target was positioned at a distance equivalent to 30% of their arm length from their trunk. The distal target was positioned at a distance equivalent to 100% of their arm length from their trunk. Participants were instructed to maintain their arm motions within the horizontal plane of movement at their relative shoulder height. In order to maintain their arm movements within this plane, a racket was attached to the frame of the two targets and placed under the participant's elbow to serve as a spatial reference throughout the task. The racket was placed far enough away from the elbow and body so that its presence did not affect the natural trunk motions. Participants moved their arm back and forth between the proximal and distal targets (starting from the proximal target), touching both gently with their index finger. The repetitive movements were carried out at a rhythm of one movement per second (2 seconds for the full cycle). This rhythm

was achieved by matching the sounds produced by the targets with the sounds produced by a metronome.

At the end of each 30-second block, the participants were asked to identify their rate of perceived exertion (RPE) in the neck/shoulder area using the modified Borg CR-10 scale [31]. Data collected during the first 30-second block was referred to No-fatigue (NF) data, while data collected during the last 30-second block was referred to as Fatigue-terminal (FT) data. The test was terminated upon occurrence of any of the following: participants reported a perceived exertion of 8 units or greater in the neck/shoulder region on the Borg CR10 scale [31], they could no longer maintain the appropriate movement rhythm of 0.5Hz, they could no longer maintain their arm elevated throughout the task (i.e. elbow touched the racket) or they surpassed a task duration of 30 minutes [28, 32]. Subjects were not informed of these stoppage criteria.

The EMG activity was measured at a sampling rate of 1000 Hz using the Telemyo 900 system (Noraxon, USA). Immediately after the pre-fatigue LTT and PPT measurements and before the RPT, pre-gelled bipolar Ag-AgCl surface electrodes (Ambu, Denmark; 1 cm diameter; 3 cm interelectrode distance) were placed over the following sites according to SENIAM recommendations: UT—the midpoint between the C7 vertebra and the angle of the acromion, AD—2 cm below the lateral third of the clavicle, BB–the midpoint on the anterior part of the upper arm, over the muscle belly, TA–the superior third of the contralateral tibialis anterior [33]. A ground electrode was placed on the spinous process of C7. The surface electrodes were oriented parallel to the muscle fibers and positioned with a 3-cm center-to-center distance. Before the surface electrodes were placed, the skin overlying the target muscles was shaved and cleaned with rubbing alcohol.

The LTT and PPT data were collected from sites just proximal from the EMG electrodes for each muscle. When recording LTT and PPT (always in that order), participants sat comfortably in a height-adjustable chair with their knees and hips flexed 90˚ and their eyes closed. Their dominant arm rested passively on a table with their forearm supinated (palm-up), their elbow extended and their shoulder in 90˚ of elevation and 40˚ of horizontal abduction. This posture was chosen as it was close to the average arm position during the RPT. Moreover, as the arm was elevated during the sensory testing procedure, it prevented fatigue recovery by restricting blood flow. Indeed, the recovery of normal sensations is delayed in such position, in comparison to a neutral arm position (unpublished observations). To measure LTT, we used a kit of twenty nylon Semmes-Weinstein monofilaments (Touch-Test™ Sensory Evaluator, North Coast Medical Inc.). Each monofilament has a different thickness, ranging from 1.65 to 6.65 on a logarithmic scale. The force necessary to make each monofilament buckle when applied at an angle of 90˚ to the surface of the skin ranges from 0.008g to 300g [34]. Starting with the thinnest, the monofilaments were first applied once in an ascending order until the participant could detect the stimulation. Then, the monofilaments were applied once in a descending order until the detection failed. Finally, a last series of stimulations was performed in an ascending order. The LTT corresponded to the first monofilament that was perceived by the participant in this final ascending phase. Following LTT measurements (which lasted around 2–3 minutes), PPT measures were taken with an electronic pressure algometer with a probe size of 1 cm$^2$ (Somedic production AB, Sweden). Pressure was applied perpendicularly to the surface of the skin with a constant increase rate of 40KPa/s. When the participant felt that the sensation of innocuous pressure changed to a sensation of painful pressure, they pushed a button on a hand-held device connected to the algometer to record their threshold. For each location, the PPT was the average of three trials performed with resting periods of 15 seconds in between. The order in which the four sites were tested was randomized and remained constant for the NF and FT assessments.

## Data analysis

EMG signals were band-pass filtered (zero-lag Butterworth, $2^{nd}$ order, [10–500 Hz]). A cross-correlation function was also used to remove heartbeats from EMG signals [35]. Data recorded during the NF and FT were partitioned into the forward and backward arm movements for each cycle, and only the forward movements were used for the analysis. The root-mean-square (RMS) and the median frequency (MDF) values were calculated and averaged over all NF and FT forward movements. Each variable was tested with a two-way (Time [repeated measure: NF vs FT] x Sex [between-subject factor: men vs women]) general estimation equation for a gamma distribution with log links **(Main objective)**. Effect size (ES) statistics with confidence intervals were computed with Hentschket et al.'s *Measures of effect size* Matlab toolbox [36]. Hedges' g were computed for Sex comparisons. Time effects ES were computed using the mdbysd function as it is a within-subject factor. Confidence intervals were computed using bootstrapping, with 10 000 iterations [37, 38]. ES were qualitatively interpreted as large (ES > 0.8), moderate (0.8 > ES > 0.5) or small/absent (ES < 0.5) as suggested by Cohen [37]. To assess the relationship between signs of local muscle fatigue and sensory inhibition/sensitisation, Pearson's correlation coefficients were computed between normalized changes ($\frac{fatigue-prefatigue}{prefatigue}$) in EMG (RMS, MDF) and sensory variables (PPT, LTT) for each muscle **(Objective 2)**. Similar correlations were computed between the PPT and the LTT **(Objective 3)**. Figures were generated with the gramm Matlab toolbox [39]. Statistical analyses were computed with SPSS 23 (IBM, USA) and Matlab 2019b (The Mathworks, USA). The level of significance of statistical analyses was set to $p \leq 0.05$. Standard deviations are presented in the text.

## Results

### Time to fatigue (Borg-8)

On average, women performed the task for 8.7 (SD: 3.4) minutes whereas men performed the task for 7.5 (SD: 3.2) minutes until they reached the first task termination criterion (i.e. in each case, reporting a score of 8 on the Borg CR-10 scale). There was no significant difference in time to fatigue between men and women ($t_{28} = 0.983$, p = 0.334).

### Muscle activity

Given that the EMG amplitude was not normalized relative to each participant's maximal voluntary activation, only the main effect of Time was considered for the RMS variables. For all upper limb muscles, main effects of Time were observed for the RMS variable, with the EMG amplitude slightly increasing with RPT-induced fatigue (Fig 2A–2C). The TA RMS did not change with RPT-induced fatigue (Fig 2D).

A large and significant decrease in EMG MDF with RPT-induced fatigue was observed in the AD (Fig 2E) muscle while the decrease was moderate for the BB (Fig 2F) muscle. No main effects of Sex or interactions were observed for the EMG MDF.

### Pressure pain and light touch sensitivity

The PPT increased slightly between NF and FT trials over the AD (Fig 3B) and the BB (Fig 3C) muscles. No significant Time effects were observed for the UT (Fig 3A) and the contralateral TA (Fig 3D). Additionally, moderate main effects of Sex were observed for the AD (Fig 3B) and the BB (Fig 3C), with women showing a greater pressure pain sensitivity than men. No interactions were observed for the PPT.

As for the LTT, moderate to large main effects of Time, indicating a decrease in light touch sensitivity with RPT-induced fatigue, were observed for all muscles, including the contralateral TA (Fig 3E–3H). Moderate to large main effects of Sex were also observed for the UT (Fig 3E)

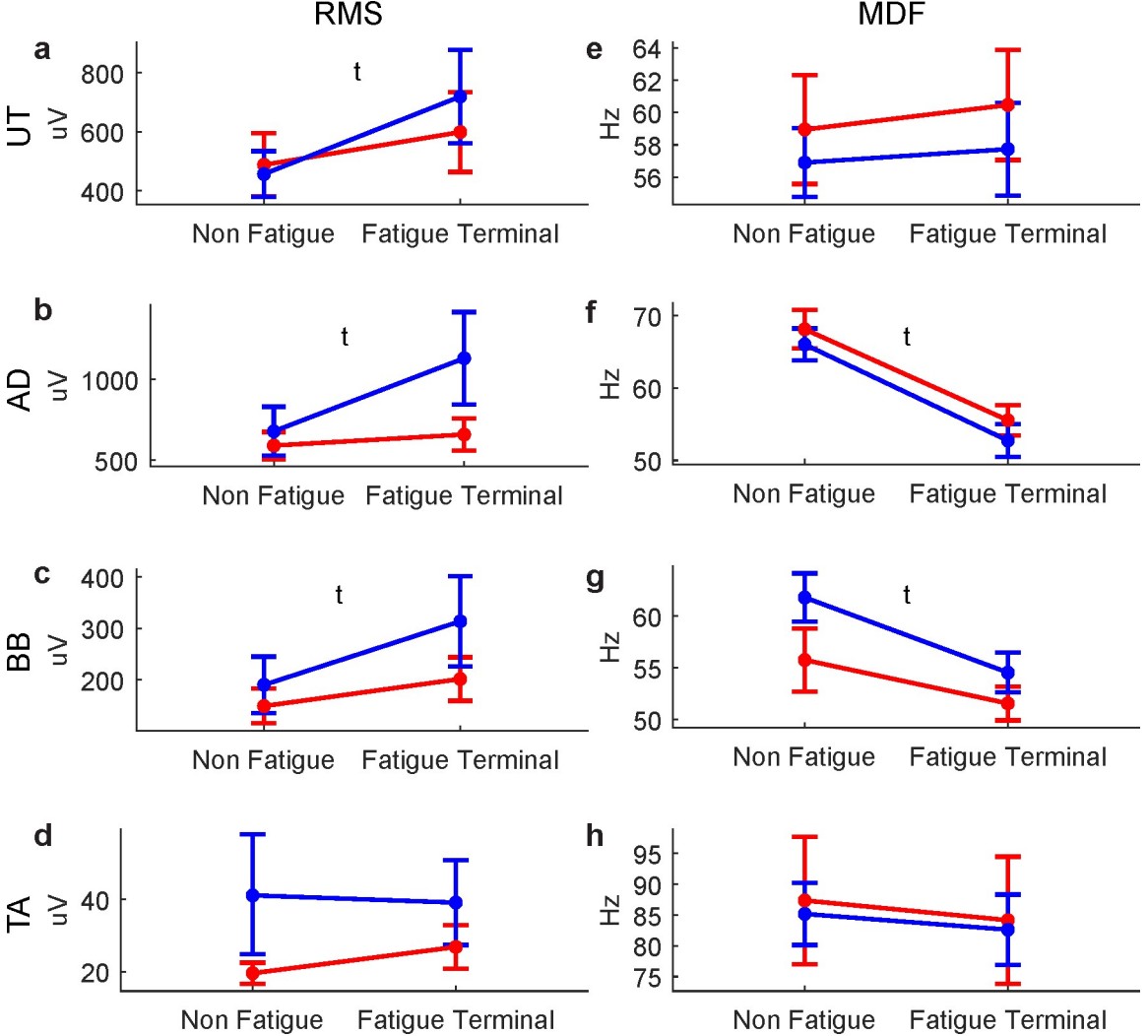

**Fig 2. Effects of fatigue and sex on motor variables.** The left panels present EMG root mean square (RMS) for the upper trapezius (UT, a), anterior deltoid (AD, b), biceps brachii (BB, c) and tibialis anterior (TA, d) muscles. The right panels present the EMG median frequency (MDF) for the UT, AD, BB and TA. s: main effect of sex, t: main effect of time, s x t: interaction (absent). Women's and men's data are represented in red and blue, respectively.

and the TA (Fig 3H) muscles, with a trend for the AD (Fig 3F). In these muscles, women had a lower LTT than men. No Sex x Time interaction was observed for the LTT.

Detailed results of statistical analyses are presented in Table 1 (p values) and Fig 4 (ES).

## Relationships between changes in EMG, PPT and LTT

Normalized changes in PPT did not correlate with changes in EMG signs of local muscle fatigue (Table 2). As for light touch sensitivity, only the UT RMS changes were weakly and positively correlated with the UT LTT changes. No correlations were found between the PPT and LTT for any muscle.

## Discussion

The current study is the first to assess concurrently the sex-specific motor as well as noxious and innocuous sensory effects of a repetitive dynamic multi-joint upper limb task. Upper limb

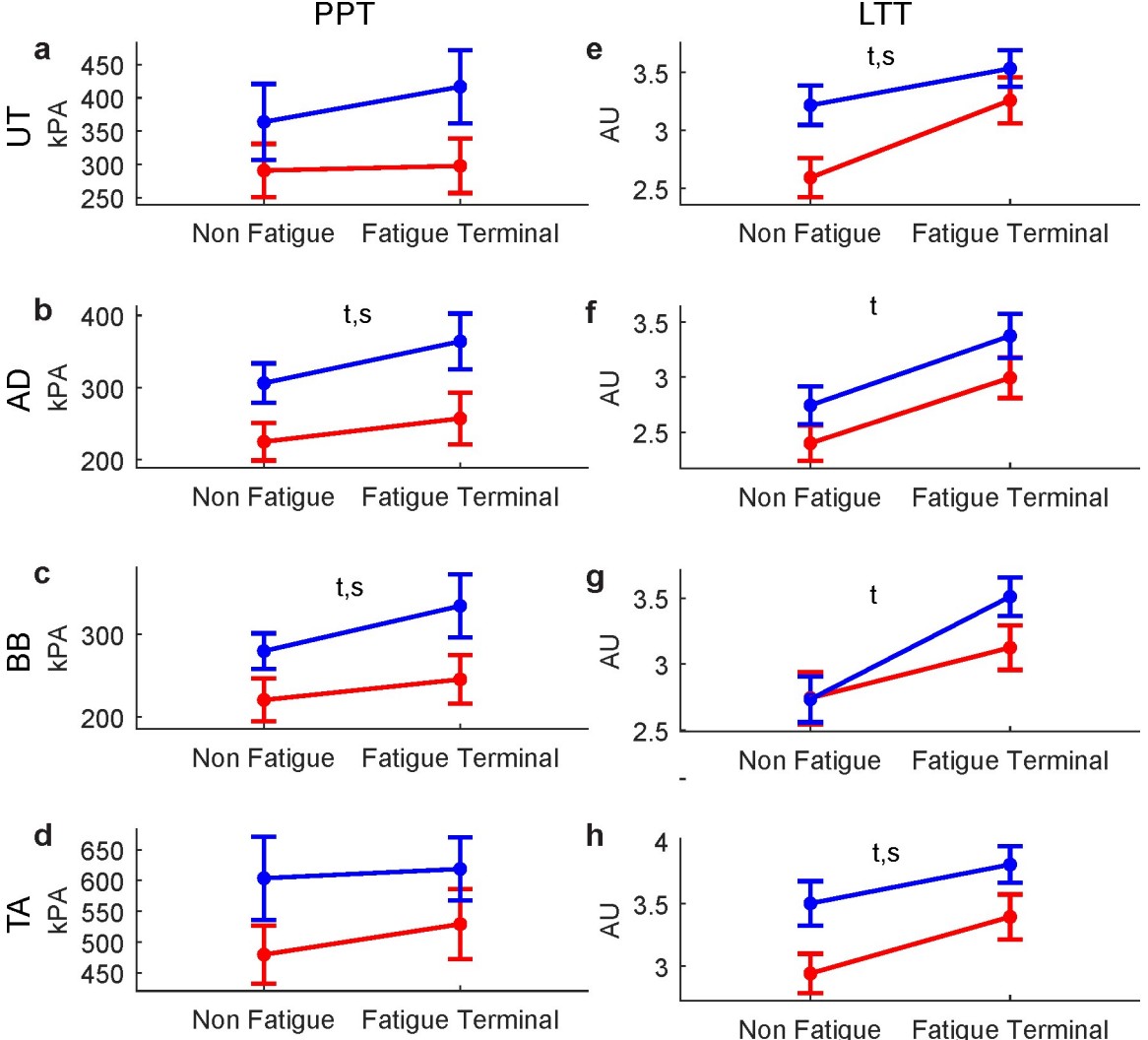

**Fig 3. Effects of fatigue and sex on sensory variables.** The left panels present the pressure point threshold (PPT) tested over the upper trapezius (UT, a), anterior deltoid (AD, b), biceps brachii (BB, c) and tibialis anterior (TA, d) muscles. The right panels present the light touch threshold (LTT) for the UT, AD, BB and TA. s: main effect of Sex, t: main effect of Time, s x t: interaction (not present). Women's and men's data are represented in red and blue, respectively.

movements repeated until reaching the previously used fatigue criteria led to an increase in the PPT over the AD and the BB muscles, consistent with the EIH phenomenon. However, EIH was not observed in the UT nor in the contralateral TA muscle. Interestingly, a significant decrease in the MDF associated with an increase in the RMS, a common sign of local muscle fatigue [10], was also only observed in the AD and the BB muscles. Although this observation suggests a relationship between local muscle fatigue and EIH, this was not supported by the correlation analyses. The LTT increased after the RPT in all assessed muscle. However, there were no sex differences in any of the measures.

## Weak muscle-dependent exercise induced hypoalgesia during the RPT

The RPT induced some hypoalgesia, as measured using pressure stimuli over the AD and the BB, but not over the UT and the contralateral TA muscles. A vast body of literature exists on

**Table 1. P values of statistical analyses regarding the effects of Sex and Time on motor and sensory variables.**

| | EMG RMS | EMG MDF | PPT | LTT |
|---|---|---|---|---|
| **Anterior deltoid** | Time:0.002* | Time: < 0.001* | Time: 0.002* | Time: < 0.001* |
| | | Sex: 0.317 | Sex: 0.027* | Sex: 0.061 |
| | | Time x Sex: 0.747 | Time x Sex: 0.694 | Time x Sex: 0.889 |
| **Biceps brachii** | Time: < 0.001* | Time: < 0.001* | Time: 0.006* | Time: < 0.001* |
| | | Sex: 0.101 | Sex: 0.049* | Sex: 0.399 |
| | | Time x Sex: 0.386 | Time x Sex: 0.493 | Time x Sex: 0.178 |
| **Upper trapezius** | Time: < 0.001* | Time: 0.288 | Time: 0.071 | Time: < 0.001* |
| | | Sex: 0.540 | Sex: 0.135 | Sex: 0.023* |
| | | Time x Sex: 0.770 | Time x Sex: 0.203 | Time x Sex: 0.103 |
| **Contralateral tibialis anterior** | Time: 0.223 | Time: 0.244 | Time: 0.096 | Time: 0.007* |
| | | Sex: 0.863 | Sex: 0.143 | Sex: 0.005* |
| | | Time x Sex: 0.906 | Time x Sex: 0.318 | Time x Sex: 0.492 |

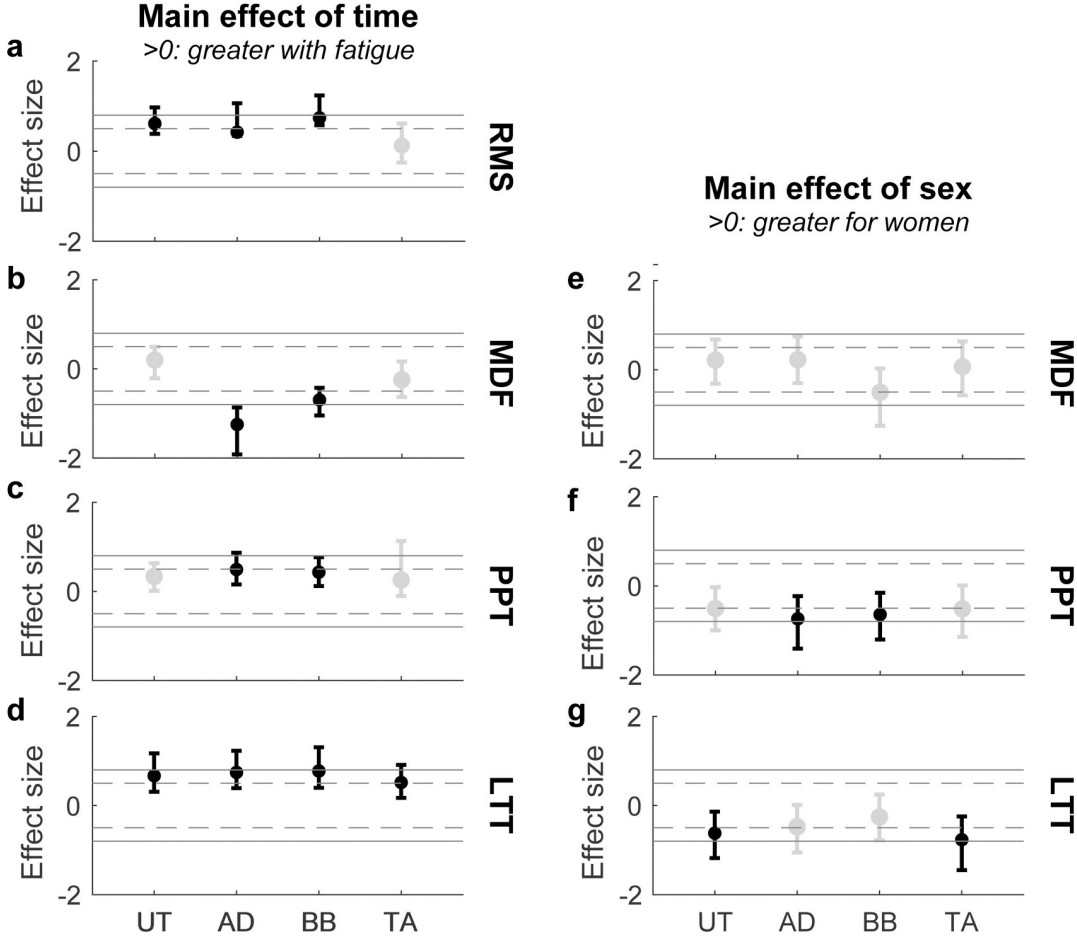

**Fig 4. Summary of results for motor and sensory variables.** Time and Sex related effect sizes for the EMG root mean square (RMS: a), EMG median frequency (MDF: b, e), pressure point threshold (PPT: c, f) and light touch threshold (LTT: d, g) variables. Dashed and full grey lines indicate thresholds for moderate (Cohen's D = ± 0.5) and high (Cohen's D = ± 0.8) effect sizes, respectively. Black and gray markers indicate significant and non-significant main effects, respectively. Error bars represent 95% confidence intervals.

**Table 2. Results of correlation analyses.**

|  | LTT | EMG RMS | EMG MDF |
|---|---|---|---|
| **PPT** | AD: r = -0.099; p = 0.602 | AD: r = 0.066; p = 0.737 | AD: r = -0.035; p = 0.859 |
|  | BB: r = 0.165; p = 0.382 | BB: r = -0.082; p = 0.686 | BB: r = 0.057; p = 0.859 |
|  | UT: r = -0.055; p = 0.773 | UT: r = 0.066; p = 0.737 | UT: r = -0.251; p = 0.198 |
|  | TA: r = -0.227; p = 0.254 | TA: r = -0.093; p = 0.672 | TA: r = 0.037; p = 0.867 |
| **LTT** |  | AD: r = 0.037; p = 0.850 | AD: r = -0.217; p = 0.268 |
|  |  | BB: r = 0.148; p = 0.461 | BB: r = -0.034; p = 0.866 |
|  |  | UT: r = 0.388*; p = 0.041* | UT: r = 0.254; p = 0.192 |
|  |  | TA: r = -0.262; p = 0.227 | TA: r = -0.274; p = 0.205 |

EIH during whole-body aerobic exercises and during localized isometric and dynamic motor tasks (reviewed in [8]). In those conditions, EIH is observed in the exercised muscles, as well as in distant ones. This contrasts with the localised hypoalgesia we observed after the RPT. The fact that no changes in the PPT were observed over the UT is surprising given its role as a scapula stabilizer when the humerus is maintained elevated [40]. Previous literature suggests that the differing roles of the UT (postural/isotonic) and of the AD/BB (dynamic) is unlikely to fully explain differences in the EIH between these muscles, as hypoalgesia has been observed after both isometric and dynamic contractions [8]. Even over the AD and the BB (prime movers during the RPT), the increase in PPT after the RPT was smaller (ES ≤ 0.36) than comparable values found in the literature ([8]; isometric: ES range = 0.27 to 2.56, weighted-average ES = 1.05; dynamic: ES range = 0.74 to 0.99, weighted-average ES = 0.83). According to Naugle et al. (2012) [8], we would expect larger EIH given the long duration (>5 minutes) and the low intensity of muscle contractions (AD: 15.1% MVC, BB: 7.7% MVC, UT: 12.6% MVC, *unpublished data from* [28]). Still, it should be noted that only two studies reviewed by Naugle et al. (2012) [8] assessed EIH following a dynamic contraction [41, 42], and that a recent study showed only modest EIH following dynamic contractions [18]. Furthermore, most studies reviewed in Naugle et al. (2012) [8] assessed EIH following motor tasks performed in a relatively neutral position. During the RPT, the participants need to maintain their arm elevated, which is a risk factor for nsMSD [3]. It is likely that the joint configuration in which exercise is performed may modulate the impact of motor activity on pain sensitivity. Indeed, a recent study from our group showed that the PPT was unchanged or even decreased (indicating hypersensitivity rather than EIH) following a sustained dexterity task performed with the arm in elevation [43].

## Lack of sex differences in EIH

The weak EIH observed with the RPT was similar for men and women in our study. This contrasts with findings from Lemley et al. (2016) [18] who observed that repeated maximal velocity elbow flexions induced a slight hypoalgesia to pressure applied on the fingertip, but only in women. Differences in task or pain induction method characteristics may contribute to these different results. Five out of the 7 studies that showed sex differences in EIH defined the PPT as the time necessary for a constant pressure applied on the fingertip to be perceived as painful [17, 18, 21, 44, 45]. This method differs from the one we employed in several aspects, including the site of the painful stimulation (muscle belly vs fingertip), its duration (≈ 10 seconds vs ≈ 30 seconds) and its intensity (≈ 300 KPA vs ≈10 N). It is likely that the fingertip pressure stimulation is more sensitive to central sensitisation and inhibitory mechanisms given the greater involvement of temporal summation of sensory inputs in this PPT definition and its remote

location relative to the exercise site. One could speculate that sex differences in EIH mostly depend on these central pain modulation mechanisms and could therefore not be detected in the current study because of the PPT protocol used. In line with this hypothesis, Gajsar et al. (2017) [46] found a greater EIH in women following isometric trunk extension when assessed in a remote arm muscle, but not in the exercised trunk and leg muscles. The effects of dynamic upper limb tasks such as the RPT on other experimental pain procedures should be explored in future research. Aside from this, the only study evaluating sex differences in shoulder muscles did not find any interaction between sex and sustained motor activity on PPT [43].

## Relationships between muscle fatigue and EIH

Some authors hypothesized that processes involved in muscle fatigue may contribute to EIH [46]. This is partly supported by our results, as the only muscles in which EIH was observed were also the only two muscles where an increase in EMG RMS occurred concurrently with a decrease in MDF, a classical sign of fatigue [10]. However, correlation analyses did not reveal the expected relationship between EMG and EIH variables. The EMG signs of fatigue result from peripheral and central changes in the neuromotor apparatus [10, 47, 48]. As for EIH, it involves several neurotransmitters regulating pain at multiple levels of the neural axis as well as the anti-inflammatory, immune and autonomous systems [14, 49]. The lack of linear relationship between EIH and the EMG changes may be related to the highly multidimensional nature of these two phenomena. An alternative hypothesis is that fatigue and EIH are actually not directly related. Of note, a previous study also failed to show correlations between the decrease in maximal voluntary force production capacity, an indicator of performance fatigability, and EIH following repeated dynamic elbow flexions or knee extensions [18].

## Effects of the RPT on light touch sensitivity

We evaluated the LTT before and after the RPT to test whether the effects of the motor task on sensory perception were specific to nociceptive inputs; they were not. Indeed, we observed an increase in LTT after the RPT over all assessed muscles, including the contralateral TA. Moreover, the effect sizes for the exercise-induced increase in LTT were all greater than for the PPT. Although the RPT mostly involves upper limb muscles, previous research showed considerable whole-body movements leading to large antero-posterior postural sway during this task, especially when upper limb fatigue has developed [28]. It is therefore possible that the changes in LTT observed over all assessed upper and lower limb muscles reflect localized effects of each segment's movements on sensory capacities. It is unlikely that this inhibition is related to the sensory gating process resulting from the efferent copy cancelling out the sensory effects of movement, since the PPT and the LTT were measured when the participants were resting [50]. The widespread decrease in light touch sensitivity with exercise may therefore reflect some peripheral or central effects leading to a relatively longstanding sensory inhibition. The many chemical changes in the peripheral, spinal and supraspinal interstitial space potentially involved in EIH may also affect the innocuous sensory sensitivity [49]. However, no linear correlations were apparent between EIH and the LTT inhibition in our data, suggesting that these two phenomena are not tightly related.

## Clinical relevance

The present study is the first, to our knowledge, to assess concurrently muscle fatigue, EIH and non-noxious sensitivity before and after a multi-joint dynamic task. The RPT is more similar to real life activities than single joint isometric or dynamic contractions, and its motor consequences have been intensively studied in men and women [22, 28]. Nevertheless, the current

study concerns the acute sensory effects of physical activity in healthy individuals. Results cannot be extrapolated to people living with musculoskeletal disorders, as EIH is often perturbed in such population [14]. Furthermore, the long-term effects of exercise on the sensory system may differ from the acute effects observed in our study. On one hand, people performing repetitive upper limb movements in their job are at increased risk of developing musculoskeletal disorders [4]. On the other hand, animal and human studies showed that regular physical activity can have protective effects against the development of disabilities [13]. It is likely that the balance between activity and rest opportunities as well as the characteristics of the physical activity performed influence its effects on sensory functions [5]. Further studies assessing the relationships between an individual's acute responses to physical activities (e.g. EIH) and the long-term risk of developing musculoskeletal disorders would be of great interest.

## Limitations

As in previous studies, the RPT was stopped when participants reached a RPE $\geq$ 8/10 [28, 43], which may represent a limitation of the study design. Using another criterion based on, for instance, actual task failure [51] or a predetermined % decrease in maximal strength [52], could modify the participants' cumulative workload and affect the obtained results. It is, however, important to note that while RPE ratings assess perceived fatigability, performance measures are not as objective as one could think. Indeed, performance limitations during a fatiguing task results from the interaction between perceived fatiguability and fatigue-related physiological changes [53]. The relative contribution of subjective and objective phenomena to performance decreases may vary between tasks and individuals. Therefore, it is unlikely that any of the above-mentioned stopping criteria can be considered as a gold standard. Still, it would be interesting to assess the impacts of the stopping criteria on sensory changes induced by a fatiguing task. A more straightforward approach would be to use a fixed motor task duration to ensure that the cumulative work performed by men and women is the same. Having said this, while many studies showed that women could maintain a static contraction for a longer time than men (reviewed in [23]), no such difference in time to fatigue is observable during the RPT [22]. The same is true for other measures of fatigability such as maximal force production capacity and heart rate, both tested previously [28]. However, previous studies showed some subtle differences in motor adaptations to the RPT between men and women [22, 24]. It is therefore possible that the achieved motor task was slightly different between men and women. In the current study, the lack of kinematic data and the fact that the EMG amplitude was not normalized limits our ability to systematically attribute fatigue effects to specific aspects of men and women's motor behavior. Furthermore, women in our sample were shorter and (non significantly) lighter than men, as expected [54]. The absolute workload and the linear (but not angular) reaching distance and speed was therefore probably larger for men than for women. Future studies should compare men and women matched for their anthropometric variables, in order to isolate specific mechanisms underlying the effects of sex, or include these variables as confounding factors given the sample size is sufficient. Other personal characteristics (e.g. fitness, history of resistance training and sports participation) should also be considered. Another limitation of the study is that only the PPT was used to assess EIH. EIH and sex differences in pain sensitivity are influenced by the experimental pain model assessed [45, 55]. Different results could be observed if pain induced by heat or chemical stimuli was assessed or if the pain intensity or tolerance was evaluated rather than the pain threshold. A similar limitation can be applied to the fatigue assessment, as it was solely assessed using EMG RMS and MDF variables. Other variables could have been chosen from EMG [10] and other signals [56, 57]. Future studies could also use electrically induced muscle contractions,

triggered at rest or superimposed to a MVC, to assess fatigue in the muscle and central nervous system. While this is a limitation for the assessment of the relationship between fatigue indicators, EIH and LTT, it does not affect the validity of our primary objective: the assessment of sex differences in EIH. Finally, in the current study, as in the complete Quantitative Sensory Testing protocol [58], LTT was always tested before PPT. We cannot exclude the possibility that the increases in LTT were larger and more widespread than the observed EIH because it was tested earlier after the end of the motor task.

## Conclusion

This study extends the literature on sex-specific EIH by using a multi-joint dynamic task designed to mimic occupational activities. EIH was smaller than expected compared to literature on single-joint isometric and dynamic contractions, or on high intensity aerobic activities. No sex differences were observed. We conclude that sex differences in the pain response to fatiguing motor activities are unlikely to contribute to women's greater prevalence of nsMSD.

## Supporting information

**S1 Database.**
(CSV)

## Author Contributions

**Conceptualization:** Jason Bouffard, Zachary Weber, Julie N. Côté.

**Data curation:** Jason Bouffard, Zachary Weber.

**Formal analysis:** Jason Bouffard.

**Funding acquisition:** Julie N. Côté.

**Investigation:** Zachary Weber, Lyndsey Pearsall, Kim Emery.

**Methodology:** Jason Bouffard, Zachary Weber, Lyndsey Pearsall, Kim Emery, Julie N. Côté.

**Project administration:** Kim Emery, Julie N. Côté.

**Resources:** Julie N. Côté.

**Supervision:** Kim Emery, Julie N. Côté.

**Validation:** Jason Bouffard, Zachary Weber, Lyndsey Pearsall, Kim Emery, Julie N. Côté.

**Visualization:** Jason Bouffard.

**Writing – original draft:** Jason Bouffard, Zachary Weber.

**Writing – review & editing:** Jason Bouffard, Zachary Weber, Lyndsey Pearsall, Kim Emery, Julie N. Côté.

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
