## [Decision Letter · Decision Letter 0]

2 Jul 2020

PONE-D-20-16305

Sex differences in the effects of fatigue on local and distal sensory and pain perception during a repetitive pointing task.

PLOS ONE

Dear Dr. Bouffard,

Thank you for submitting your manuscript to PLOS ONE. After careful consideration, we feel that it has merit but does not fully meet PLOS ONE’s publication criteria as it currently stands. Therefore, we invite you to submit a revised version of the manuscript that addresses the points raised during the review process.

We look forward to receiving your revised manuscript.

Kind regards,

Nizam Uddin Ahamed, PhD

Academic Editor

PLOS ONE

Journal Requirements:

2. Please amend either the abstract on the online submission form (via Edit Submission) or the abstract in the manuscript so that they are identical.

Additional Editor Comments (if provided):

Reviewers' comments:

Reviewer's Responses to Questions

**Comments to the Author**

1. Is the manuscript technically sound, and do the data support the conclusions?

Reviewer #1: Partly

Reviewer #2: Yes

Reviewer #3: Partly

Reviewer #4: Yes

2. Has the statistical analysis been performed appropriately and rigorously? 

Reviewer #1: Yes

Reviewer #2: Yes

Reviewer #3: Yes

Reviewer #4: No

3. Have the authors made all data underlying the findings in their manuscript fully available?

Reviewer #1: Yes

Reviewer #2: Yes

Reviewer #3: Yes

Reviewer #4: Yes

4. Is the manuscript presented in an intelligible fashion and written in standard English?

Reviewer #1: Yes

Reviewer #2: Yes

Reviewer #3: Yes

Reviewer #4: Yes

5. Review Comments to the Author

Reviewer #1: General comments

Bouffard and co-authors aimed at investigating sex differences in pain and sensitivity following a fatiguing upper limb task. Their rationale/justification was based on greater prevalence of neck-shoulder musculoskeletal disorders being more frequent in women compared to men in jobs that require repetitive upper limb movements. They subjected 15 men and 15 women to a repetitive pointing task until perceived fatigue of 8 on the 0-10 Borg scale. Prior and after the task, they assessed pressure pain and light touch sensitivity on 4 muscles sites, 3 of those related to the task and 1 distal and not task related. EMG of these muscles was recorded during the task. They have found no sex differences in pain or sensitivity. The authors concluded that no sex differences on pain and sensitivity were observed after a repetitive pointing task and suggested that their results could not explain sex differences seen in repetitive work scenarios. Overall, this is a well written study and apparently was rigorously conducted. However, I have some major concerns.

Specific comments

1. My main concern is related to the task. The repetitive pointing task was very well described in your methodology, however, the use of a Borg scale to indicate fatigue and cessation of exercise is very subjective and concerning in my opinion. You cannot guarantee that your subjects, between- and within-sex, exercised at a similar relative volume, which plays an important role on muscle fatigue. Another way to look at this is to measure task-related muscle fatigue; this was considered by the authors but with some flaws (see comment #2 below). Did the authors consider other stoppage criteria? Task failure?

2. Another main concern is related to muscle fatigue. Firstly, the authors did not define, or at least did not specify which definition of fatigue they considered while collecting data. It is known that muscular fatigue can be interpreted in many ways. Moreover, the use of EMG to determine fatigue of a muscle is somewhat contradictory. EMG is used to assess electrical activity, which can be high even if the muscle is fatigued. Changes in EMG activity do not represent/indicate muscle fatigue. A proper measure of fatigue would be a baseline vs. post-task muscle force output comparison at a known and constant intensity stimulus, for example, electrical or magnetic stimulation.

3. The authors did not provide anthropometrical description of their sample. Ideally, strength-matched men and women would provide a better base of comparison. In addition, a simple measurement with a bioimpedance scale (fat and lean mass) would have provided important information to interpret your results. Moreover, it was not indicated whether subjects provided signed consent to participate and if ethics were conformed to the Declaration of Helsinki.

4. As a reader, the way the authors present their results is somewhat hard to follow. For example, it would be easier if important data (such as some of your results and those in your database file) were shown in 1 or 2 tables, pre vs. post task. Moreover, your figures do not have legends as to the different colors used. The use of too much acronyms makes it harder to follow as well. I suggest reviewing which ones are really necessary and removing the others that are not used too often within text.

5. Did the authors consider a mid-/long-term intervention as to simulate a longer fatiguing exposure? In other words, do you think a single and acute exposure to the task in question would elicit musculoskeletal responses similarly to those observed in repetitive task workers? Your rationale was based on this matter, but I do not see how your methodology fit in here (acute vs. long-term exposure).

6. I suggest a careful review on your references; for example, you made reference to a study on Semmes-Weinstein filaments when talking about a cross-correlation function used to remove ECG ‘noise’ from EMG data (Ref #29, line 169).

7. I also have some minor comments: a) in some instances you mention ‘many’ studies and refer to only 1 (lines 56-57); even being a review, it still sounds weird to the reader; b) line 84: lower limb (“central” ??); c) line 93: regarding ‘right-handed’, was this an inclusion criteria? If not, please state; d) line 103, you mention PPT and LTT measurements were ‘immediately following’ RPT, my question is how long did it take each muscle to be assessed? Maybe your results would be different if the order of muscles assessment were different? Have the authors considered this?

Reviewer #2: Can the authors provide a rationale for the lack of kinematic data and why EMG amplitude were not normalized in the current study?

Is n=15/group sufficient to observe an effect size?

Given the lack of sex-specific differences, the title is a bit misleading and can be modified to state lack of sex differences.

Can the authors provide a rationale for why they chose this study design which could be compared with published findings due to differences in design?

Reviewer #3: This manuscript assessed sex-related differences in electromyography, pressure pain threshold, and light-touch sensitivity thresholds of the upper trapezius, anterior deltoid, biceps brachii, and the contralateral tibialis anterior muscle for a repetitive pointing task. Results indicated no sex-related differences for any of the measures. A decrease in pain sensitivity (hypoalgesia) was not correlated with muscle fatigue for any muscle tested. The authors conclude that fatigue-related sensory factors do not, for the presented protocol, offer any explanation for why females may be more prone to upper body workplace injuries compared to males (as denoted by no sex-related differences in experimental pain mechanisms related to repetitive tasks).

As the authors note, there are significant limitations to the protocol. Primarily, the lack of normalization for EMG data; however, other limitations include: no disclosure of further participant characteristic data (i.e., weight, height, lever arm length, activity history), no randomization of timing for testing pressure pain threshold (PPT) before the light touch threshold (LTT; LTT was assessed before PPT in all cases, potentially leading to null findings given a possible quick recovery time for measures of interest), and no disclosure of any metronome adjustments for task frequency based on arm length to account for differences in distance traveled.

The findings of this study are interesting and complementary to the literature; however, a number of concerns (such as those listed above and below) should be addressed in the manuscript to clarify the results. Some of these concerns are with the design of the study (ie. Not testing PPT before LTT, not randomizing these measures, and/or not having two separate testing days to account for differences in time-from-fatigue), which, if addressed appropriately within the text, can improve the quality of the paper greatly.

Comments:

Introduction

The general flow of the introduction is present, but the transition between concepts is choppy. This makes it difficult to follow the storyline projected by the authors such that a strong defence for their hypotheses might be presented. The authors highlight the importance of investigating sex-related differences in nsMSD in the first paragraph and follow this in the next paragraph by a discussion of the link between fatigue, pain, and MSD. The main objective highlights the desire to assess light touch, pain pressure thresholds, and EMG measures of fatigue; however, the former two are not clearly differentiated in the introduction. This should be addressed.

Additionally, why was it important to differentiate between local vs distal effects; that is, why would it be assumed that LTT and PPT would have distal effects yet EMG measures of fatigue would not? Please highlight this briefly in the introduction.

- Line 58-60: Stating “EIH” followed by “induced hypoalgesia” comes across as repetitive. Consider rewording to improve clarity.

- Line 62: The authors mentioned a number of neurotransmitters and that these contribute to EIH – do they all enhance effects of EIH or do some neurotransmitters work to oppose it? Please include the directionality of their contribution.

Methods

- Line 93: Please indicate whether this is SD or SEM for the participant ages. Additionally, please separate the average ages for males and females.

- Please include more descriptive participant characteristics for each sex; i.e., weight, height, forearm/arm length, general activity history (were these moderately active individuals, athletes, sedentary, or a mixed group? Were individuals screened for a history of resistance/weight training?).

- Were all participants screened for usage of antidepressant/anxiolytic medications? Had any taken pain medications prior to testing?

- Were female participants screened for contraceptive medication usage and/or tested at a certain point during their menstrual cycle? if not, please highlight the rationale for this decision in the manuscript.

- Indicate whether all participants gave written informed consent prior to starting the experimental protocol.

- Was the timing of the metronome adjusted for participants with longer arms, given they would have had to travel a longer distance between targets? If not, what was the rationale for keeping these the same? If you took measures of arm length, you can assess for a sex difference to see if perhaps one sex had to travel greater distances than another sex.

- Lines 110-111: Was stance regulated for participants with different heights? If so, how.

- Line 130-131: How often was RPE assessed by the experimenter? Was this assessment consistent throughout the protocol and across participants?

- Line 137: Please indicate the diameter of the recording surface of the electrodes.

- Line 170-171: Why did was the backward arm motion not assessed?

Data analysis:

- What statistical software was used for your assessments?

- Please indicate your pre-determined p-value for setting statistical significance.

Results:

- Line 221: Please remove the word “trends” as there is no statistical significance for the values presented.

Discussion:

- If PPT was assessed after LTT for all participants post-fatigue, is it possible that the amount of time between cessation of the protocol to the beginning of assessing PPT was enough to allow for any potential effect of sex to recover? If you had tested this measure immediately following fatigue instead of testing LTT first, perhaps there would be a different story. Can you speculate on this?

- Line 316: should read “previous study” not “previous research”

- Line 323-324: Again, it may be possible that these effects were different based on how time-sensitive they are to when the fatiguing task ceased. In that case, a larger effect should be expected for LTT.

Strengths/Limitations:

- Line 339: This is the first time the term “whole-body dynamic task” has shown up in reference to the repetitive pointing task. This task cannot be considered whole-body given that it primarily works the upper limb and has no involvement from the lower limb, other than what is required for quiet standing, and therefore is not comparable to true whole-body dynamic exercise such as running or cycling. Please revise the terminology.

Conclusion:

- Line 354: Please see note above about the term “whole-body dynamic task”

Reviewer #4: The paper is interesting, well written and the problem is well stated. I have some minor concerns and/or suggestions for improvement:

1. the authors have used RMS and MDF of EMG for assessment of the muscle fatigue. However, recent works use different measures [RCIM19]. The authors should mention novel measures, and they should motivate the reasons why they have used EMG RMS and MDF. For istance, why joint torque is not included?

[RCIM19] Peternel, Luka, et al. "A selective muscle fatigue management approach to ergonomic human-robot co-manipulation." Robotics and Computer-Integrated Manufacturing 58 (2019): 69-79.

2. the authors should include some pictures of the people performing the task with the EMG placement on the body.

3. the authors have performed a statistical analysis using the effect size. They should include the hypotheses on the distribution of data to evaluate the effect size (normality and homogeneity of variance). However, does it make sense to perform the effect size analysis if there is not a significative differences in the data (line 200)?

4. the author should include the significance level of the p-value. Without it, it is difficult to give an interpretation on the p-values equal to 0.096 and 0.071 (line 222)

5. based on the achieved results, the authors should include a discussion on possible future works related to understanding of the real causes of the WMSD in the women.

6. PLOS authors have the option to publish the peer review history of their article (what does this mean?). If published, this will include your full peer review and any attached files.

Reviewer #1: No

Reviewer #2: No

Reviewer #3: No

Reviewer #4: No

---

## [Author Response · Author response to Decision Letter 0]

18 Aug 2020

Dear reviewers,

I would first like to thank each of you for the rigorous review made of the submitted manuscript entitled: “Similar effects of fatigue on local and remote sensory and pain perception during a repetitive pointing task in men and women.”. Many important comments were made by each reviewer which help us improve our manuscript. We are please to submit you a revised version of our manuscript in which we addressed most comments. Several changes were made to the manuscript including: a restructuration of the Introduction section, the addition of a Figure (i.e. experimental set-up) and two Tables (i.e. statistical results) and the inclusion of confidence intervals on Figure 4. Some limitations were also discussed in the discussion section as well as a Clinical relevance section. We are confident that the quality and potential impact of the paper were greatly improved, thanks to the insightful comments of the reviewers.

You will find in an attached documents, answers to each reviewer’s comments along with modifications that were made to the manuscript.

Sincerely,

Jason Bouffard, OT Ph.D.

---

## [Decision Letter · Decision Letter 1]

8 Oct 2020

PONE-D-20-16305R1

Similar effects of fatigue on local and remote sensory and pain perception during a repetitive pointing task in men and women.

PLOS ONE

Dear Dr. Bouffard,

Thank you for submitting your manuscript to PLOS ONE. After careful consideration, we feel that it has merit but does not fully meet PLOS ONE’s publication criteria as it currently stands. Therefore, we invite you to submit a revised version of the manuscript that addresses the points raised during the review process.

We look forward to receiving your revised manuscript.

Kind regards,

Nizam Uddin Ahamed, PhD

Academic Editor

PLOS ONE

Reviewers' comments:

Reviewer's Responses to Questions

**Comments to the Author**

1. If the authors have adequately addressed your comments raised in a previous round of review and you feel that this manuscript is now acceptable for publication, you may indicate that here to bypass the “Comments to the Author” section, enter your conflict of interest statement in the “Confidential to Editor” section, and submit your "Accept" recommendation.

Reviewer #1: (No Response)

Reviewer #2: All comments have been addressed

Reviewer #3: (No Response)

2. Is the manuscript technically sound, and do the data support the conclusions?

Reviewer #1: Partly

Reviewer #2: Yes

Reviewer #3: Yes

3. Has the statistical analysis been performed appropriately and rigorously? 

Reviewer #1: Yes

Reviewer #2: Yes

Reviewer #3: Yes

4. Have the authors made all data underlying the findings in their manuscript fully available?

Reviewer #1: Yes

Reviewer #2: Yes

Reviewer #3: Yes

5. Is the manuscript presented in an intelligible fashion and written in standard English?

Reviewer #1: Yes

Reviewer #2: Yes

Reviewer #3: Yes

6. Review Comments to the Author

Reviewer #1: Although an important improvement was made to the manuscript, I'm still not convinced on your responses to my #1 and #2 comments.

Regarding comment #1: I still think the use of RPE scores as stoppage criterion is a limitation and a flaw in your design. I agree with your response but I still believe a better approach to this would be matching the cumulative work performed by your male and female subjects. This should be listed as a limitation.

Regarding comment #2: I agree it is difficult to assess fatigue on a multi-joint task, involving multiple muscles. As you mentioned, EMG is a good way to infer signs of fatigue, however, I still believe it has its limitations and you could have used another way to assess fatigue. Possibly, of all muscles tested, you could have picked one that is the most 'fatigable' in the literature, and tested for fatigue in the way I mention (baseline vs post-task, force output after a non-volitional contraction - electrical/magnetic stimulus).

Other than that, I commend the authors for the big improvement made to the manuscript.

Reviewer #2: I have no further comments. The authors have addressed all concerns that were raised in the previous round and provided new data as well as analysis.

Reviewer #3: The authors have adequately addressed my main comments. My only major remaining concern are the grammatical errors throughout the manuscript (syntax, etc) that will impair the ability of a reader to understand certain statements. The authors should have the manuscript thoroughly reviewed by a native English speaker with an academic writing background.

Minor comments:

L12: "shorter duration contractions" - do you mean high intensity contractions with a short duration? This sentence reads as though these shorter duration contractions are at a low/moderate contraction intensity, which would imply that the resulting fatigue should be less than the long duration contractions. Revise accordingly.

L20-25: Grammatical errors within this sentence make it very difficult to navigate. Please revise. I have refrained from making further comments about language, as I believe this paper would greatly benefit from professional editorial services.

Methodology: The authors noted in their review response that participants were not screened for physical activity history. I would strongly encourage future studies incorporate at least a basic measure of physical activity screening. Resistance training history or professional sports (i.e., baseball, rugby, basketball) can heavily influence fatiguability of upper limb muscles.

7. PLOS authors have the option to publish the peer review history of their article (what does this mean?). If published, this will include your full peer review and any attached files.

Reviewer #1: No

Reviewer #2: No

Reviewer #3: No

---

## [Author Response · Author response to Decision Letter 1]

20 Nov 2020

Dear reviewers,

I would first like to thank you for the high quality comments made in the current and previous review rounds for our manuscript entitled: “Similar effects of fatigue induced by a repetitive pointing task on local and remote light touch and pain perception in men and women”. We are pleased to submit you a revised version of our manuscript in which we addressed the remaining comments you formulated. The manuscript was entirely reviewed to correct grammatical errors as highlighted by reviewer 3. Some information was also added in the Limitations section of the discussion.

You will find answers to each reviewer’s comments along with modifications that were made to the manuscript in the Response to reviewers document.

Sincerely,

Jason Bouffard, OT Ph.D.

---

## [Decision Letter · Decision Letter 2]

8 Dec 2020

Similar effects of fatigue induced by a repetitive pointing task on local and remote light touch and pain perception in men and women

PONE-D-20-16305R2

Dear Dr. Bouffard,

We’re pleased to inform you that your manuscript has been judged scientifically suitable for publication and will be formally accepted for publication once it meets all outstanding technical requirements.

Kind regards,

Nizam Uddin Ahamed, PhD

Academic Editor

PLOS ONE

Additional Editor Comments (optional):

Reviewers' comments:

Reviewer's Responses to Questions

**Comments to the Author**

1. If the authors have adequately addressed your comments raised in a previous round of review and you feel that this manuscript is now acceptable for publication, you may indicate that here to bypass the “Comments to the Author” section, enter your conflict of interest statement in the “Confidential to Editor” section, and submit your "Accept" recommendation.

Reviewer #1: All comments have been addressed

Reviewer #2: All comments have been addressed

Reviewer #3: All comments have been addressed

2. Is the manuscript technically sound, and do the data support the conclusions?

Reviewer #1: Yes

Reviewer #2: Yes

Reviewer #3: Yes

3. Has the statistical analysis been performed appropriately and rigorously? 

Reviewer #1: Yes

Reviewer #2: Yes

Reviewer #3: Yes

4. Have the authors made all data underlying the findings in their manuscript fully available?

Reviewer #1: Yes

Reviewer #2: Yes

Reviewer #3: Yes

5. Is the manuscript presented in an intelligible fashion and written in standard English?

Reviewer #1: Yes

Reviewer #2: Yes

Reviewer #3: Yes

6. Review Comments to the Author

Reviewer #1: All suggestions and comments were properly addressed. I commend and congratulate the authors for their work.

Reviewer #2: In this second round of revisions, the authors have addressed all concerns raised and I have no further comments or concerns.

Reviewer #3: Just a side comment: perhaps consider exchanging the usage of women/men to female/male, as the latter suggests biological sex and the former is considered a social construct.

7. PLOS authors have the option to publish the peer review history of their article (what does this mean?). If published, this will include your full peer review and any attached files.

Reviewer #1: No

Reviewer #2: No

Reviewer #3: No

---

## [Editor Report · Acceptance letter]

10 Dec 2020

PONE-D-20-16305R2 

Similar effects of fatigue induced by a repetitive pointing task on local and remote light touch and pain perception in men and women 

Dear Dr. Bouffard:

I'm pleased to inform you that your manuscript has been deemed suitable for publication in PLOS ONE. Congratulations! Your manuscript is now with our production department. 

Kind regards, 

on behalf of

Dr. Nizam Uddin Ahamed 

Academic Editor

PLOS ONE